# Time Series Missing Imputation with Multivariate Radial Based Function Neural Network

## Abstract

Researchers have been persistently working to address the issue of missing values in time series data. While numerous models have been proposed, they often come with challenges related to assumptions about the model or data and the instability of deep learning. This paper introduces an imputation model that can be utilized without explicit assumptions. Our imputation model is based on the Radial Basis Function (RBF) and learns local information from timestamps to create a continuous function. Additionally, we incorporate time gaps to facilitate learning information considering the missing terms of missing values. We name this model the Missing Imputation Multivariate RBFNN (MIM-RBFNN). However, MIM-RBFNN relies on a local information-based learning approach, which presents difficulties in utilizing temporal information. Therefore, we propose an extension called the Missing Value Imputation Recurrent Neural Network with Continuous Function (MIRNN-CF) using the continuous function generated by MIM-RBFNN. We evaluate the performance using two real-world datasets and conduct an ablation study comparing MIM-RBFNN and MIRNN-CF.

## 1 Introduction

Multivariate time series data finds extensive application across various domains such as healthcare, weather forecasting, finance, and transportation (Hsieh et al., 2011; Kaushik et al., 2020; Jia et al., 2016; Schwartz & Marcus, 1990). In healthcare, time series data are used for classifying patient outcomes, such as mortality or recovery. In contrast, they are employed to tackle regression problems related to precipitation, stock prices, and traffic volume in domains like weather, finance, and transportation. However, the collection of multivariate time series data in multiple domains is often plagued by irregular time intervals and issues like equipment failures in collective devices (García-Laencina et al., 2010). This instability leads to missing data in multivariate time series datasets, posing significant challenges for data analysis and mining (García-Laencina et al., 2015). Consequently, addressing missing data in time series data is recognized as a highly critical issue.

The strategies for addressing missing data can be broadly categorized into two main approaches: statistical methods and model-centric methods. Statistical approaches for missing value imputation commonly employ techniques such as mean, median, and the imputation of the most frequently occurring values (Zhang, 2016). These methods offer the advantage of simplicity and are relatively time-efficient. However, they may yield inaccurate results since they do not take into account the temporal aspects of time series data. In contrast, model-based missing data imputation methods have been put forward (Nelwamondo et al., 2007; Josse & Husson, 2012; Che et al., 2018). These methods introduce techniques for capturing temporal information within time series data. Nonetheless, they often rely on strong assumptions, such as presuming that the data adheres to a particular model distribution, or they incorporate information beyond what is available in multivariate time series data, such as specific location data. Consequently, these approaches may struggle to capture the data distribution adequately and may not guarantee good performance on data that deviates from the model assumptions or lacks location information.

In this study, we propose a missing data imputation model for time series data by leveraging both the Radial Basis Function Neural Network (RBFNN) and the Recurrent Neural Network (RNN).

RBFNN is a neural network that utilizes the linear combination of Radial Basis Functions (RBFs), which are nonlinear. When RBFNN is employed, it approximates data by generating approximate continuous functions that capture local information within the data (Ng et al., 2004; Yang et al., 2022). When Gaussian RBF (GRBF) is used in RBFNN, it learns the covariance structure of the data by capturing local information. Furthermore, GRBF generates smoother curves over a longer time span, allowing it to model periodic patterns and nonlinear trends in time series data (Corani et al., 2021). To extend this capability of learning covariance structures to multivariate time series data, we introduce the Missing Imputation Multivariate RBFNN (MIM-RBFNN) with respect to the time stamp ($t$). However, because RBFNN relies on local information to learn covariance structures, it encounters challenges in utilizing temporal information effectively. Therefore, we additionally propose the Missing Value Imputation Recurrent Neural Network with Continuous Function (MIRNN-CF), which leverages the continuous functions generated by MIM-RBFNN.

## 2 RELATED WORK

Efficient management of missing data is crucial for smooth time series data analysis. In recent years, numerous endeavors have put forth strategies to address this issue. The most straightforward approach involves eliminating instances that contain missing data (Kaiser, 2014). However, this method can introduce analytical complexities as the rate of missing data increases. In addition to deletion, various statistical imputation techniques, such as replacing missing values with the mean value (Allison, 2001), imputing them with the most common value (Titterington, 1985), and completing the dataset by using the last observed valid value (Cook et al., 2004), have been proposed. One of the benefits of these methods is that they allow for the utilization of a complete dataset without requiring data deletion.

Recent studies have introduced imputation methods based on machine learning. These machine learning-based approaches can be categorized into non-Neural and Neural Network-based methods. Non-Neural Network-based methods encompass techniques like maximum likelihood Expectation-Maximization (EM) based imputation (Nelwamondo et al., 2007), K-Nearest Neighbor (KNN) based imputation (Josse & Husson, 2012), and Matrix Factorization (MF) based imputation (Koren et al., 2009). The EM imputation algorithm predicts missing values by leveraging model parameters. The K-Nearest Neighbor (KNN) based imputation method imputes missing values by calculating the mean value of k-neighbor samples surrounding the missing value. In contrast, the Matrix Factorization (MF) based imputation method utilizes low-rank matrices $U$ and $V$ to perform imputation on the incomplete matrix. However, these approaches rely on strong assumptions regarding missing data (Cao et al., 2018; Du et al., 2023).

Recently, there has been a growing trend in the development of missing value imputation methods utilizing Recurrent Neural Networks (RNNs). Yoon et al. (2018) introduced the M-RNN (Multi-directional Recurrent Neural Network) framework based on RNNs (Yoon et al., 2018). M-RNN operates bidirectionally, similar to a bi-directional RNN (bi-RNN), to perform imputation for missing data. Also, some researchers have proposed studies employing a variant of RNN known as the Gate Recurrent Unit (GRU). Che et al. (2018) introduced GRU-D, a GRU-based model, which effectively captures temporal dependencies by integrating masking and time interval-based missing patterns into the GRU cell, demonstrating improved predictions by utilizing missing patterns. However, the practical usability of GRU-D may be restrained when there is no clear inherent correlation between the missing patterns in the dataset and the prediction task. Cao et al. (2018) presented BRITS, a model based on bidirectional recurrent dynamics. BRITS also utilizes masking and time intervals, achieving strong performance in missing value imputation. Furthermore, there have been proposals for combining generative adversarial networks (GANs) with models for imputing missing values in time series data. Luo et al. (2018) introduced GRUI for imputation, proposing a GRUI-GAN model that incorporates a generator and discriminator based on GRUI, effectively accounting for temporal dependencies without ignoring them. Moreover, researchers have been steadily investigating various deep learning models such as NAOMI (Liu et al., 2019) for imputing long-term missing data and SAITS (Du et al., 2023) based on self-attention. These approaches have consistently driven advancements in the state-of-the-art for time series imputation. Nevertheless, it is important to note that deep learning-based imputation models, functioning as autoregressive models, face challenges such as compounding errors (Liu et al., 2019; Venkatraman et al., 2015), difficulties in training generative models, non-convergence, and mode collapse (Wu et al., 2020; Salimans et al., 2016).

## 3 RBF FOR MISSING VALUE IMPUTATION OF TIME SERIES DATA

To address the challenge of missing values in time series data, we introduce the MIM-RBFNN model. It aims to solve the problem by creating an appropriate continuous function for each time series to handle missing value imputation. The RBF neural network is renowned for effectively approximating any nonlinear function and is often referred to as a universal function approximator (Yu et al., 2011). In this section, we comprehensively explain the structure and learning techniques employed in MIM-RBFNN, which is grounded in RBF approximation. Before delving into the specifics of MIM-RBFNN, we define RBF and introduce the concept of nonlinear approximation using RBF.

### 3.0.1 RADIAL BASIS FUNCTION (RBF)

Radial Basis Function (RBF), denoted as $\varphi$, is a basis function whose value depends on the distance from a specified point, often called the "center." It can be mathematically expressed as $\varphi(x) = \hat{\varphi}(|x|)$. RBFs are real-valued functions as they are defined based on real numbers. Typically, instead of the origin, a fixed point $c$ is chosen as the center, and the RBF is redefined as $\varphi(x) = \hat{\varphi}(|x - c|)$. While the Euclidean distance is commonly used for distance calculation, alternative distance metrics can also be employed.

### 3.0.2 NON-LINEAR APPROXIMATION USING RBF

The summation of RBFs is employed to construct an approximation function that suits the given data. The RBF approximation method is considered continuous due to its reliance on the distance between two points within a continuous function (RBF). Suppose we represent the RBF as $\varphi(|x - c_i|)$ and the approximate function as $f(x) = \sum_{i=1}^{n} w_i \varphi(|x - c_i|)$, where $n$ denotes the number of RBFs, and $w_i$ signifies the associated weights. The RBF approximation method offers several compelling advantages. First, it is computationally efficient because it primarily focuses on approximating the local characteristics near the center $c_i$ (Wettschereck & Dietterich, 1991). Second, it enhances the smoothness of data fitting by utilizing multiple RBFs (Carr et al., 2001).

### 3.1 MULTIVARIATE-RBFNN FOR MISSING DATA IMPUTATION

We utilize non-linear approximation using Radial Basis Functions (RBFs) ($\varphi$) to address the challenge of missing values in multivariate time series data. We extend the RBFNN to a multivariate RBFNN to effectively handle missing values in multivariate time series. Furthermore, to accommodate missing time intervals, we propose using Gaussian RBFs ($\varphi_k(x) = \exp\left(\frac{-(x - c_k)^2}{2\sigma_i^2}\right)$). This model is called the Missing value Imputation Multivariate RBFNN (MIM-RBFNN).

The MIM-RBFNN aims to impute missing values by generating a suitable continuous function for the input data. For this purpose, MIM-RBFNN employs Radial Basis Functions (RBFs), with each RBF taking a timestamp ($t$) as input ($\varphi_k(t)$) and fitting the corresponding value ($X_t^m$) at that timestamp. Here, $x_t^m$ represents the $m$-th feature value of the variable $X$ at timestamp $t$. To model periodic patterns and nonlinear trends in time series data, we utilize Gaussian Radial Basis Functions (GRBFs). GRBFs that take a timestamp ($t$) as input capture the local information of $t$. Our model has parameters $c_k$ and $\sigma_k$ for GRBFs, which are trained to create an approximate continuous function tailored to the target time series data. Additionally, to apply this approach to multivariate time series, we train different linear weights $w_k^m$ to create continuous functions for each variable.

We train the center vector $c_k$ to determine the optimal center vector for target time series values based on the input $t$. Each GRBF captures the local characteristics of input variables near $c_k$ (Chen et al., 2013). If we consider $H$ as the diagonal covariance matrix, the GRBF $\varphi_k(t) = \exp\left(\frac{-(t - c_k)^2}{2\sigma_i^2}\right)$ is equivalent to $\exp(-\frac{1}{2}(t - c_k)^{-1} H_k(t - c_k))$. As a result, the RBFNN, by traning $\sigma_k$, discovers the optimal diagonal covariance $H_k$ for target based on input time (Chen et al., 2013). Our MIM-RBFNN tracks the optimal center vector $c_k$ and diagonal covariance matrix $H_k$ to capture the local characteristics of input timestamp $t$, and we extend this approach to multivariate time series data.

$CF^m = \sum_k w_k^m e^{-\frac{(t - c_k)^2}{\sigma_k}}$ illustrates the continuous function generated by MIM-RBFNN for time series data ($X^m$). In MIM-RBFNN, multivariate time series data share the same GRBFs while being

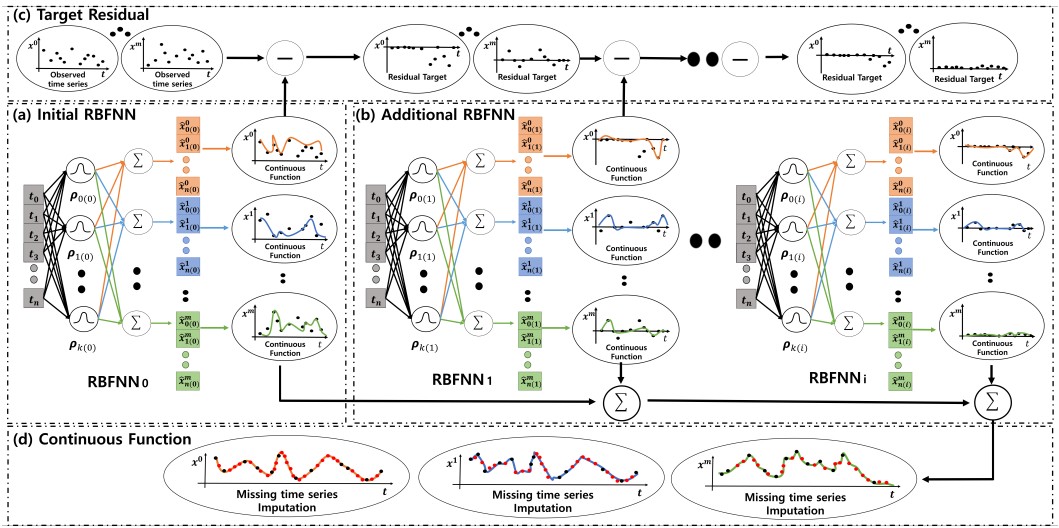

Figure 1: MIM-RBFNN architecture. RBFNN$_i$ is the $i$-th RBFNN with $\varphi_{k(i)}$ as its $k$-th RBF.

trained to track common $c_k$ and $H_k$. Additionally, we compute different linear weights ($w_k^m$) for each variable ($X^m$) to create appropriate approximation functions (continuous functions). Therefore, by sharing the diagonal covariance structure of multivariate time series, we generate a continuous function ($CF^m$) for imputing missing data. An ablation study regarding the GRBF sharing aspect of MIM-RBFNN can be found in Appendix C.

Figure 1 illustrates the architecture of MIM-RBFNN, which comprises four distinct processes: (a) the Initial RBFNN Process, (b) the Additional RBFNN Process, (c) the Target Residual Process, and (d) the Missing Time Series Imputation Process. The challenge is determining the number of RBFs employed in the linear combination to fit the target time series variable for RBF approximation adequately. However, the precise quantity of RBFs needed for each variable remains uncertain. We utilize processes (a) and (b) to tackle this issue. Both (a) and (b) role to ascertain the requisite number of RBFs for approximating the target data, i.e., the observed time series data. As depicted in (a), the Initial RBFNN Process entails a learning phase dedicated to constructing a continuous function using the initial RBFNN$_0$. Subsequently, we calculate the approximation error of the continuous function generated by the initial RBFNN$_0$. If the fitting error surpasses the loss threshold (MAPE 5%), the Additional RBFNN Process is executed in (b), where additional RBFs are trained. The Additional RBFNN process in (b) involves training an additional RBFNN with the same architecture as the initial RBFNN$_0$. However, during the Additional RBFNN process, if we were to use the target time series as is, it would necessitate retraining the previously trained RBFNN, which would lead to an increase in the number of parameters that ought to be trained, resulting in longer training times (Rasley et al., 2020). Such a scenario can escalate the complexity of the training process and introduce confusion (Tian et al., 2015). Therefore, as illustrated in (c), we exclusively train RBFs of the additional RBFNN with the target residual process. In (c), the target residual process calculates the residual of the Continuous Function generated in (a) from the initial observed time series data to update the target data. The updated target data becomes the first target in (b). The first additional RBFNN$_1$ in (b) utilizes the updated target data to create a continuous function. Subsequently, the target data is updated using the continuous function generated by the first additional RBFNN$_1$ and the residual from the updated target data. This updated target data serves as the target data for the second additional RBFNN$_2$. Process (c) continues until the Additional RBFNN Process is completed. Finally, MIM-RBFNN combines all the GRBFs trained in (a) and (b) to generate a Continuous Function for multivariate time series in (d) and imputes missing values.

## 3.2 INITIAL PARAMETERS OF GRBFS

Unlike MLP networks that typically initialize parameters randomly, RBFNN requires the initial state of parameters to be explicitly specified (Yu et al., 2011). In this section, we introduce the strategies for initializing initial parameters in MIM-RBFNN.

**Initial Centers.** MIM-RBFNN trains the RBFNN through the processes depicted in Figure 1(a) and (b). We employ distinct methods to allocate initial centers for the RBFNNs in (a) and (b). GRBFs have their highest value at the center. Consequently, the RBFNN in the (a) process is assigned initial centers based on timestamps with higher values in the target multivariate time series data. In the case of (b), the target time series is updated to the Residual Target by (c). Hence, the initial centers in the (b) process are assigned to timestamps with higher values in the Residual Target.

**Initial Weights.** The RBFNN weights determine each GRBF's magnitude, signifying the center value of a symmetrical Gaussian distribution curve for the GRBF function (Shaukat et al., 2021). We assign initial centers to times with the highest errors, indicating that the times around the initial centers have low values. Therefore, to assign the initial centers at the center of the symmetrical Gaussian distribution curve of the initial centers ($c_k = t$), we assign the target value ($X_t^m$) of each time series to the initial weights ($w_k^m$) of each time series.

**Initial Sigmas.** The $\sigma$ in the RBFNN represents the width of each GRBF, which in turn defines the receptive field of the GRBF. Moreover, training the $\sigma$ aims to uncover the optimal diagonal covariance matrix $H$ based on the input time stamp's target time series values (Chen et al., 2013). Since our MIM-RBFNN takes time $t$ as its input, the receptive fields of the GRBFs are closely tied to the local information of the time stamp $t$ that MIM-RBFNN utilizes. Consequently, we incorporate a time gap to facilitate the learning of local information by each Gaussian RBF, taking into account missing values in the surrounding time points (Cao et al., 2018; Yoon et al., 2017).

$$\delta_n^m = \begin{cases} 0 & \text{if } i = 0 \\ t_n - t_{n-1} & \text{if } m_{n-1}^m = 1 \ \& \ n > 0 \\ \delta_{n-1}^m - t_n + t_{n-1} & \text{if } m_{n-1}^m = 0 \ \& \ n > 0 \end{cases} \tag{1}$$

Equation 1 represents the time gap. This time gap calculation measures the time difference between the current timestamp and the timestamps without missing values. Since the $\sigma$s of the GRBFs define their receptive ranges, we introduce the time gap as a factor into $\sigma$ to enable each GRBF to account for missing values. Furthermore, to facilitate the joint learning of the covariance structure in each multivariate time series, we initialize the $\sigma$ for each time series as the mean of their corresponding time gaps. The ablation study concerning the initial $\sigma$ can be reviewed in Appendix B. Besides, a more comprehensive description of the backpropagation algorithm for MIM-RBFNN parameters is found in Appendix A.

## 4 TEMPORAL INFORMATION WITH RECURRENT NEURAL NETWORK

In this section, we describe the covariance structure trained with MIM-RBFNN and the imputation model utilizing RNN. As previously explained, MIM-RBFNN leverages local information from multivariate time series data to perform missing value imputation. However, the approximation achieved using RBFNN relies on learning local information based on observed values to create a continuous function. Furthermore, since our MIM-RBFNN also learns based on local information, its utilization of temporal information is relatively limited compared to Recurrent Neural Networks (RNNs) (Liu et al., 2020; Gao & Er, 2005). This limitation becomes more pronounced as the length of the missing value term increases, potentially impacting imputation performance. Therefore, we propose an imputation model that combines the continuous function generated by MIM-RBFNN with the bidirectional recurrent dynamics temporal information learned by RNNs.

### 4.1 BIDIRECTIONAL RECURRENT WITH RBF CONTINUOUS FUNCTION

Figure 2 illustrates the architecture that combines the continuous function generated by MIM-RBFNN with bidirectional recurrent dynamics. This model is called the "Missing value Imputation Recurrent Neural Network with Continuous Function" (MIRNN-CF). The input data for MIRNN-CF includes Continuous Function data, a time gap matrix, a Mask matrix, and incomplete time series data. MIRNN-CF employs an RNN that utilizes feature-based estimation loss, historical-based estimation loss, and consistency loss for bidirectional RNN training, similar to previous studies (Cao et al., 2018; Miao et al., 2021). Besides, we propose a Continuous-concatenate estimation loss that combines the continuous function with RNN predictions.

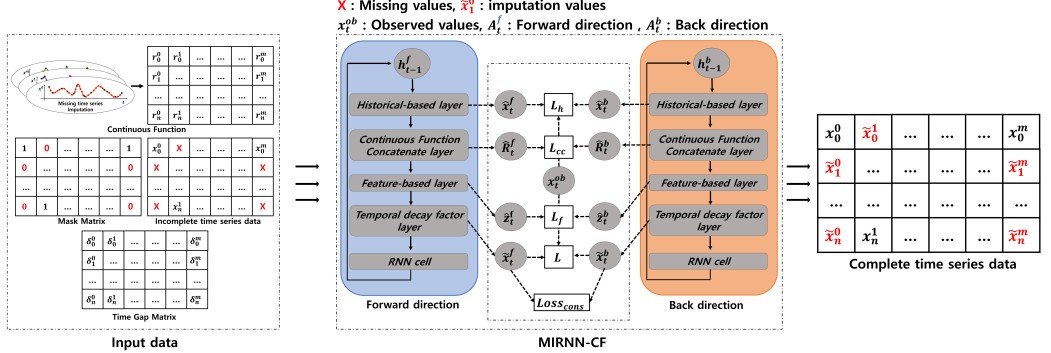

Figure 2: MIRNN-CF architecture.

## 4.2 MIRNN-CF STRUCTURE

The structure of MIRNN-CF is based on the state-of-the-art BRITS model (Cao et al., 2018) with modifications. In a standard RNN, the hidden state $h_t$ is continually updated to learn temporal information (Elman, 1990). Historical-based estimation assesses the hidden state $h_{t-1}$ with a linear layer. Equation 2 represents historical-based estimation $\hat{x}_t$.

$$\hat{x}_t = W_x h_{t-1} + b_x \tag{2}$$

$$x_t^c = (1 - m_t)\hat{x}_t + m_t x_t \tag{3}$$

Equation 3 represents the imputed complete data obtained from the historical-based estimation (Cao et al., 2018). To incorporate the covariance structure of multivariate time series data into the estimation based on the RNN's hidden state, we concatenate the Continuous Function data with the estimated $x_t^c$.

$$\hat{R}_t = W_{cf}CF_t + W_{xc}x_t^c + b_R \tag{4}$$

$$R_t^c = (1 - m_t)\hat{R}_t + m_t x_t \tag{5}$$

Equation 4 represents the regression layer concatenating historical-based estimation with Continuous Function data. In Equation 4, $CF_t$ refers to the continuous function data generated by MIM-RBFNN. Equation 5 depicts the complete data based on the estimation from Equation 4. However, incorporating the covariance structure of this continuous function data, $\hat{R}_t$, involves combining its own covariance structure, potentially limiting the utilization of information from other features. To include information from other features, we employ feature-based estimation.

$$\hat{z}_t = W_z R_t^c + b_z \tag{6}$$

Equation 6 describes the feature-based estimation. In Equation 4, we train the self-covariance. Therefore, to utilize only the information from other variables, we set the diagonal parameters of $W_z$ to zeros in Equation 6 (Cao et al., 2018). In other words, $\hat{z}_t$ in Equation 6 is a feature-based estimation that leverages the covariance structure of the Continuous Function data. Finally, we employ a temporal decay factor ($\gamma_t$) (Che et al., 2018) in Equation 7 to combine feature-based estimation with historical-based estimation. We also follow the structure of previous feature-based estimation and historical-based estimation studies to combine both estimations using Equation 8 as shown in Equation 9.

$$\gamma_t = \tau(W_r \delta_t + b_r) \tag{7}$$

$$\beta_t = \sigma(W_\beta[\gamma_\circ m_t] + b_\beta) \tag{8}$$

$$\tilde{x}_t = \beta_t \odot \hat{z}_t + (1 - \beta_t) \odot \hat{x}_t \tag{9}$$

$$\bar{x}_t = m_t \odot x_t + (1 - m_t) \odot \tilde{x}_t \tag{10}$$

MIRNN-CF operates bidirectionally through the processes mentioned above to derive $\bar{x}_t^{forward}$ and $\bar{x}_t^{back}$, taking their average as the prediction for complete data. The hidden state update in MIRNN-CF utilizes $[h_t \odot \gamma_t]$ and $[\bar{x}_t \odot m_t]$ as inputs to update the RNN cell. In other words, MIRNN-CF leverages Continuous Function data, time gap matrix, Mask matrix, and Incomplete time series data

to predict complete data $\bar{x}_t$. MIRNN-CF minimizes the combined loss function, as shown in Equation 11, which includes the feature-based estimation loss ($\mathcal{L}_f$), historical-based estimation loss ($\mathcal{L}_h$), Continuous-concatenate estimation loss ($\mathcal{L}_{cc}$), and the consistency loss ($\mathcal{L}_{cons}$).

$$
\begin{aligned}
\mathcal{L}_{\mathcal{MIRNN-CF}} = \mathcal{M} \odot \mathcal{L}(\mathcal{X}, \tilde{\mathcal{X}}) + \mathcal{M} \odot \mathcal{L}_h(\mathcal{X}, \hat{\mathcal{X}}) + \\
\mathcal{M} \odot \mathcal{L}_f(\mathcal{X}, \hat{\mathcal{Z}}) + \mathcal{M} \odot \mathcal{L}_{cc}(\mathcal{X}, \hat{\mathcal{R}}) + \mathcal{L}_{cons}(\bar{\mathcal{X}}^{forward}, \bar{\mathcal{X}}^{back})
\end{aligned}
\tag{11}
$$

## 5 EVALUATION

We evaluate the performance of our proposed models, MIM-RBFNN and MIRNN-CF, using real-world datasets. We compare their missing value imputation performance with baseline models using the mean absolute error (MAE) and mean relative error (MRE) metrics. Furthermore, we conduct an Ablation Study to compare the effectiveness of temporal information learning between MIM-RBFNN and MIRNN-CF. The experiments were conducted on an Intel Core 3.60GHz server with a Geforce RTX 3090 GPU and 64GB of RAM.

### 5.1 DATASETS AND BASELINE

**Air Quality Data.** We utilized Beijing air quality data, including PM 2.5 collected from 36 monitoring stations from May 1, 2014, to April 30, 2015 (Yi et al., 2016) to evaluate missing data imputation. It provides data with missing values, ground truth data, and the geographical information of each monitoring station. Approximately 25% of the data contains missing values, while the ground data covers around 11% of ground-truth values for the missing data. We trained the model using the observed data. We compared the imputed complete data, generated utilizing the observed information from the incomplete data, with the ground truth.

**Human Activity Data.** We also utilized the Localization Data for Person Activity dataset available from the UCI Machine Learning Repository (Cao et al., 2018; Miao et al., 2021) to assess imputation performance. This dataset captures the activities of five individuals engaged in various tasks, with each person wearing four tags (ankle left, ankle right, belt, and chest) to record the x-coordinate, y-coordinate, and z-coordinate data for each tag. These experiments were conducted five times for each person, resulting in approximately 4,000 data points, each consisting of 40 consecutive time steps. Notably, this dataset does not contain any missing values. To evaluate missing value imputation, we artificially generated missing values randomly at rates of 30%, 50%, and 80%, while the original dataset served as the ground truth.

**Baseline.** For performance comparison, we have chosen eight baseline models. We selected models that utilize temporal information less significantly, similar to MIM-RBFNN (such as Mean, k-nearest neighbor, MICE). Additionally, we included three models (M-RNN, BRITS) and a self-attention model (SAITs) to compare with RNN-based models like MIRNN-CF.

### 5.2 RESULT

Table 1 presents the missing value imputation performance on real-world datasets. We rounded the MRE values to four decimal places for both datasets, while MAE values for the air quality dataset were rounded to two decimal places. For the human activity dataset, they were rounded to four decimal places. For the experimental settings, we kept all random seeds fixed. For the air quality dataset, we used 36 consecutive time steps, a batch size of 64, and a hidden size of 64 as constants. Additionally, we generated continuous functions for the 36 monitoring stations using MIM-RBFNN. For the human activity dataset, we used 40 consecutive time steps, a batch size of 64, and a hidden size of 64 as fixed parameters. Similarly, we generated continuous functions for each person's experiment, creating four continuous functions for each person's four tags using MIM-RBFNN.

The empirical results demonstrate that both MIM-RBFNN and MIRNN-CF exhibit enhanced imputation performance compared to baseline models when applied to real-world datasets. The imputation performance was notably superior. As seen in Table 1, MIRNN-CF significantly improves imputation performance over baseline models, particularly showcasing a 30% to 50% enhancement

Table 1: Imputation performance comparison (MAE(MRE))

| Method | Air quality | Human activity | | |
|---|---|---|---|---|
| | | 30% | 50% | 80% |
| Mean | 55.5 (0.779) | 0.453 (0.274) | 0.453 (0.274) | 0.454 (0.275) |
| KNN | 29.4 (0.413) | 0.497 (0.0.301) | 0.709 (0.429) | 1.208 (0.730) |
| MF | 38.2 (0.536) | 0.465 (0.282) | 0.478 (0.290) | 0.482 (0.291) |
| MRNN | 20.5 (0.299) | 0.363 (0.213) | 0.433 (0.263) | 0.441 (0.266) |
| SAITS | 19.3 (0.281) | 0.343 (0.208) | 0.372 (0.221) | 0.423 (0.258) |
| BRITS | 13.1 (0.186) | 0.171 (0.103) | 0.287 (0.174) | 0.310 (0.188) |
| **MIM-RBFNN** | 22.1 (0.31) | 0.150 (0.091) | 0.163 (0.098) | 0.265 (0.161) |
| **MIRNN-CF** | **12.3 (0.172)** | **0.101 (0.061)** | **0.138 (0.084)** | **0.224 (0.136)** |

in the MAE metric compared to the baseline BRITS model for human activity data. We also found that MIM-RBFNN performs better than BRITS in human activity data across all missing rates. However, MIM-RBFNN does not outperform other deep learning models regarding air quality data. Due to the modest performance of MIM-RBFNN, the missing imputation performance of MIRNN-CF falls short of showing substantial improvements over baseline models for air quality data compared to human activity data. To analyze this further, we conduct a comparative analysis of MIM-RBFNN's results for both datasets.

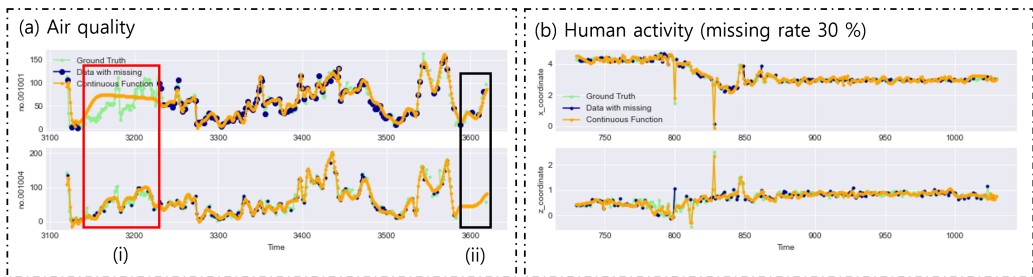

Figure 3: Comparison of continuous function

Figure 3 illustrates the continuous function generated by MIM-RBFNN. MIM-RBFNN performs better than the baseline on human activity data but falls short of the baseline on air quality data. To investigate this further, we compare the continuous functions for both datasets. Figure 3 (a) presents the continuous function for air quality data. Upon examining Figure 3 (i), it becomes evident that the continuous function struggles to perform proper imputation in long-term missing cases, primarily because no data is available to capture the local covariance structure. However, Figure 3 (ii) shows that, despite long-term missing data, the continuous function has learned the multivariate covariance structure, following a trend similar to other variables. On the other hand, human activity data was generated with a 30% random missing rate. Figure 3 (b) demonstrates that human activity data, with its shorter instances of long-term missing compared to air quality data, leverages observation data to learn the local covariance structure for imputation. In conclusion, MIM-RBFNN's utilization of temporal information is modest, making it challenging to handle imputation for long-term missing data. However, for scenarios where long-term missing data is not as prevalent, such as in Figure 3 (b), MIM-RBFNN can learn local covariance structures effectively and perform well even with substantial missing rates.

## 5.3 ABLATION STUDY OF MIRNN-CF

As previously mentioned, we identified that MIM-RBFNN faces challenges when dealing with long-term missing data. To address this issue, we proposed the inclusion of temporal information learning in MIRNN-CF. In this section, we present an ablation study to delve deeper into this topic.

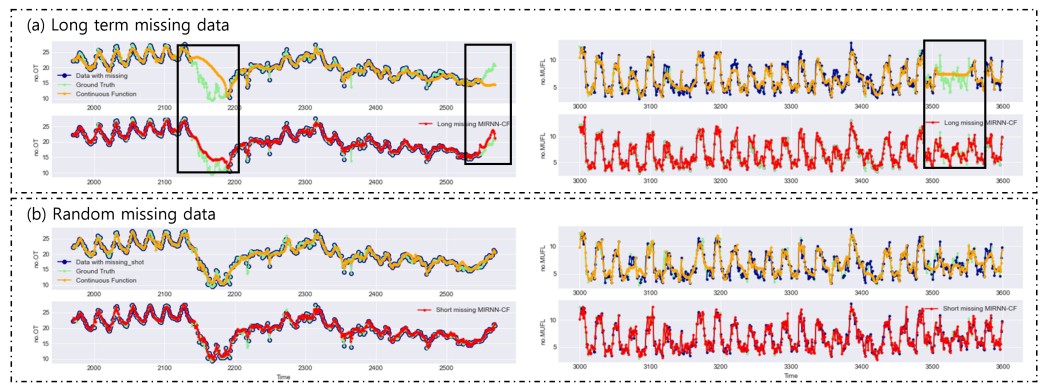

Figure 4: Comparison of continuous functions according to missing term

To validate the effectiveness of using temporal information learning in MIM-RBFNN to address long-term missing imputation challenges, we employ the Electricity Transformer Temperature (ETT) dataset available from the UCI Machine Learning Repository (Zhou et al., 2021). The ETT dataset comprises seven multivariate time series data collected hourly and 15-minute intervals from July 1, 2016, to June 26, 2018. To maintain consistency with our previous experiments, we focus on the hourly dataset, specifically the data spanning 12 months from July 1, 2016, to June 30, 2017. To generate long-term missing data, we produced long-term missing data with a missing rate of 20%, including missing terms ranging from 50 to 80, along with random missing values. In comparison, we also create random missing data with a 20% missing rate, containing short-term missing values about the temporal missing term from 1 to 8.

For the ETT dataset, the performance metrics (MAE and MRE) reveal that MIM-RBFNN yields an MAE of 1.298 (MRE of 0.226) for long-term missing data and an MAE of 0.735 (MRE of 0.129) for random missing data. These results highlight the challenges faced by MIM-RBFNN when dealing with long-term missing data, similar to what was observed in the air quality dataset. However, leveraging MIM-RBFNN to implement MIRNN-CF results in a notable performance improvement, particularly for long-term missing data, where MIRNN-CF achieves an MAE of 0.563 (MRE of 0.098). Figure 4 visually represents the ablation study's outcomes. Figure 4 (a) showcases the continuous function generated by MIM-RBFNN (represented by the orange line) for long-term missing data, along with the results obtained by MIRNN-CF utilizing this continuous function (depicted by the red line). The results in (a) exhibit challenges in generating a continuous function for long-term missing data, akin to the observations made in Figure 3. Nevertheless, MIRNN-CF, which learns temporal information, effectively performs imputation. Furthermore, Figure 4 (b) displays the continuous function for random missing data (orange line) and imputation results similar to those obtained with MIRNN-CF (red line).

## 6 CONCLUSIONS

In this study, we proposed two models for addressing missing values in multivariate time series data: MIM-RBFNN, which leverages the learning of local covariance structures using GRBF, and MIRNN-CF, a hybrid model combining continuous function generation with RNN. We demonstrated the effectiveness of MIM-RBFNN through experiments on real-world datasets. However, MIM-RBFNN primarily relies on local information to generate continuous functions, revealing challenges in learning temporal information and the complexities of long-term missing data imputation. To tackle these issues, we introduced MIRNN-CF, which utilizes MIM-RBFNN's continuous functions, as verified through experiments on air quality data and an ablation study focused on long-term missing data imputation. Nevertheless, we approached this by developing separate models for MIM-RBFNN and MIRNN-CF. As a future research direction, we plan to investigate the development of a unified model that simultaneously learns local covariance structures based on RBF's local information and temporal information through RNN.

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

## A    Back Propagation Algorithm of MIM-RBFnn

To generate a continuous function for time series data based on the time stamp $(t)$, we update the optimal parameters $(c, \sigma, w)$ of each GRBF using the Backpropagation (BP) algorithm. Each parameter is updated as follows: $W^m = W^m - lr * \frac{\partial L}{\partial W^m}$, $C = C - lr * \frac{\partial L}{\partial C}$, and $\Sigma = \Sigma - lr * \frac{\partial L}{\partial \Sigma}$, where $lr$ denotes the learning rate.

$$\varphi_k = e^{\frac{-(t-c_k)^2}{\sigma_k}}, \ F^m = \sum_{k=0} w_k^m \varphi_k \ , \ L = \sum (\frac{X^m - F^m)^2}{N} \tag{12}$$

$$\frac{\partial L}{\partial w_k^m} = \sum_k \frac{\partial L}{\partial F^m} \frac{\partial F^m}{\partial w_k^m} = \sum_k \frac{\partial L}{\partial F^m} \varphi_k \tag{13}$$

$$\frac{\partial L}{\partial c_k} = \sum_k \frac{\partial L}{\partial F^m} \frac{\partial F^m}{\partial \varphi_k} \frac{\partial \varphi_k}{\partial c_k} = \sum_m \sum_k \frac{\partial L}{\partial F^m} w_k^m \frac{\partial \varphi_k}{\partial c_k} \tag{14}$$

$$\frac{\partial L}{\partial \sigma_k} = \sum_k \frac{\partial L}{\partial F^m} \frac{\partial F^m}{\partial \varphi_k} \frac{\partial \varphi_k}{\partial \sigma_k} = \sum_m \sum_k \frac{\partial L}{\partial F_k^m} w_k^m \frac{\partial \varphi_k}{\partial \sigma_k} \tag{15}$$

Equations 13, 14, 15 represent the parameter updates of MIM-RBFNN using gradient descent. Here, $\varphi_k$ indicates the $k$-th GRBF. The parameters of each GRBF are updated based on its receptive field. In particular, each GRBF learns the covariance structure within its receptive field and returns zero for data outside its receptive field, minimizing its impact on data beyond that field. Equation 13 shows the update process for weights $(W^m)$ for each time series $(X^m)$. Equations 14 and 15 depict the update process for the centers $(c_k)$ and sigmas $(\sigma_k)$ of GRBFs, respectively, while jointly considering the covariance structure of multivariate time series data. Based on this BP algorithm, each GRBF iteratively seeks the optimal centers, weights, and sigmas for its receptive field, building upon the values from the previous update step (Wu et al., 2012).

## B    Appendix : Comparison of initial sigma in MIM-RBFNN

The parameter sigma $(\sigma)$ of RBFNN determines the width of the radial basis function, which represents the receptive field of the RBF. We assign the initial $\sigma$s of MIM-RBFNN using Equation 1 to accommodate missing values in time series data. In this section, we analyze the impact of the initial $\sigma$ assignment on time series missing value imputation. For comparative analysis, we compare two methods of assigning the initial $\sigma$ in MIM-RBFNN: one where it is assigned randomly following $\mathcal{N}(0, 1)$, and the other using Equation 1.

Table 2: Imputation performance comparison for initial $\sigma$ (MAE(MRE))

| Method | Air quality | ETT | |
|---|---|---|---|
| | | Random | Long term |
| **MIM-RBFNN** | **22.1 (0.31)** | **0.735 (0.129)** | **1.298 (0.226)** |
| MIM-RBFNN + Random | 25.9 (0.363) | 0.836 (0.147) | 1.470 (0.256) |

Table 3: Imputation performance comparison for initial $\sigma$ (MAE(MRE))

| Method | Human activity | | |
|---|---|---|---|
| | 30% | 50% | 80% |
| **MIM-RBFNN** | **0.150 (0.091)** | **0.163 (0.098)** | **0.224 (0.136)** |
| MIM-RBFNN + Random | 0.216 (0.131) | 0.276 (0.167) | 0.886 (0.536) |

Table 2 and 3 display the results of missing data imputation performance based on the random $\sigma$ and the initial $\sigma$ assignment using Equation 1. Upon reviewing Tables 2 and 3, it is evident that for all datasets, assigning the initial $\sigma$ as described in Equation 1 to accommodate missing values yields

better results compared to randomly assigning $\sigma$, which stresses the importance of considering the receptive field of RBF concerning time gaps when imputing missing values, highlighting the effectiveness of our initial $\sigma$ assignment method. Furthermore, we generated synthetic data to analyze the changes in the receptive field of MIM-RBFNN during the $\sigma$ learning process. We used Lorenz-96 data as our synthetic dataset (Karimi & Paul, 2010), creating 200 time stamps with five variables. Subsequently, we introduced random missing data, accounting for 30% of the data. In terms of imputation performance (MAE(MRE)), MIM-RBFNN (1.706 (0.418)) outperforms MIM-RBFNN + Random (2.141 (0.525)), showcasing the superior performance of MIM-RBFNN that accounts for the time gaps.

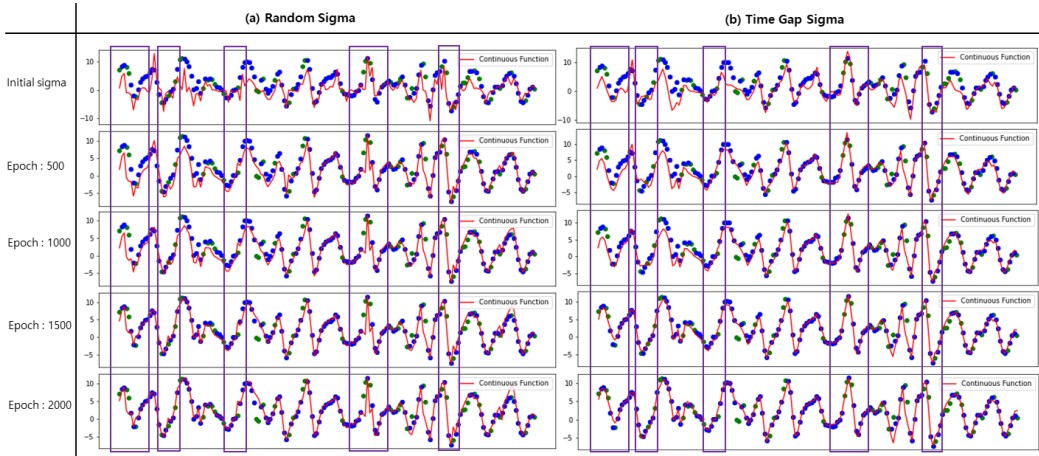

Figure 5: Comparison of continuous function learning process according to initial $\sigma$

Figure 5 illustrates the learning progress on synthetic data, where green points represent the ground truth of missing data, blue points represent observed data, and the red line represents the continuous function. When examining the violet boxes in Figure 5 (a) and (b), it becomes evident that the continuous function of MIM-RBFNN, which accounts for time gaps, is adapting to missing time points. In contrast, for MIM-RBFNN + Random, the learning process is only biased towards the observed data, emphasizing that the initial $\sigma$ of MIM-RBFNN allows it to create a continuous function using a receptive field that considers missing data.

## C  APPENDIX : COMPARISON OF MULTIVARIATE RBF AND SINGLE VARIATE RBF

We propose MIM-RBFNN, which shares the same RBF for training the diagonal covariance structure of multivariate time series data. In this section, we analyze the impact of sharing the same RBF on imputing missing values in multivariate time series data. To conduct this analysis, we compare RBFNN and MIM-RBFNN, which use separate RBFs instead of sharing the same one. We refer to the model that uses separate RBFs as "Missing data imputation Single RBFNN" (MIS-RBFNN). MIS-RBFNN is identical to MIM-RBFNN except that it uses separate RBFs for each variable.

Table 4: Imputation performance comparison between MIM-RBFNN and MIS-RBFNN (MAE(MRE))

| Method | Air quality | ETT | |
|---|---|---|---|
| | | Random | Long term |
| **MIM-RBFNN** | **22.1 (0.310)** | **0.735 (0.129)** | **1.298 (0.226)** |
| MIS-RBFNN | 26.8 (0.377) | 0.751 (0.132) | 1.507 (0.262) |

Table 5: Imputation performance comparison for initial $\sigma$ (MAE(MRE))

| Method | Human activity | | |
|---|---|---|---|
| | 30% | 50% | 80% |
| **MIM-RBFNN** | **0.150 (0.091)** | **0.163 (0.098)** | **0.224 (0.136)** |
| MIS-RBFNN | 0.182 (0.110) | 0.203 (0.123) | 0.286 (0.173) |

MIM-RBFNN employs the same set of shared RBFs for all variables and learns all variables within a single model, utilizing distinct weights ($w^m$) for each variable. However, MIS-RBFNN employs separate and unique RBFs for each variable, necessitating one MIS-RBFNN model per variable. Consequently, for the Air quality dataset, we utilized 36 distinct MIS-RBFNN models, while for the ETT dataset, we employed seven different MIS-RBFNN models to generate continuous functions. Likewise, for the Human activity dataset, following the approach used in previous experiments, we used MIS-RBFNN to create continuous functions for each individual's experiment with four tags. The results of missing data imputation for MIM-RBFNN and MIS-RBFNN are found in Table 4 and 5. Upon reviewing Table 4 and 5, it becomes evident that MIM-RBFNN consistently outperforms MIS-RBFNN across all datasets, which emphasizes the fact that MIM-RBFNN, through the use of shared RBFs, learns multivariate diagonal covariance structures, thereby enhancing its missing value imputation performance.

## D  APPENDIX : COMPARISON OF SINGLE VARIATE RBF AND INITIAL SIGMA

Finally, we consolidate the results in Appendix B and C and analyze the impact of the initial $\sigma$ in MIS-RBFNN.

Table 6: Imputation performance comparison (MAE(MRE))

| Method | Air quality | ETT | |
|---|---|---|---|
| | | Random | Long term |
| **MIM-RBFNN** | **22.1 (0.310)** | **0.735 (0.129)** | **1.298 (0.226)** |
| MIM-RBFNN + Random | 25.9 (0.363) | 0.836 (0.147) | 1.470 (0.256) |
| MIS-RBFNN | 26.8 (0.377) | 0.751 (0.132) | 1.507 (0.262) |
| MIS-RBFNN + Random | 32.5 (0.457) | 0.947 (0.167) | 1.682 (0.293) |

Table 7: Imputation performance comparison (MAE(MRE))

| Method | Human activity | | |
|---|---|---|---|
| | 30% | 50% | 80% |
| **MIM-RBFNN** | **0.150 (0.091)** | **0.163 (0.098)** | **0.224 (0.136)** |
| MIM-RBFNN + Random | 0.216 (0.131) | 0.276 (0.167) | 0.886 (0.536) |
| MIS-RBFNN | 0.182 (0.110) | 0.203 (0.123) | 0.286 (0.173) |
| MIS-RBFNN + Random | 0.359 (0.217) | 0.577 (0.349) | 1.295 (0.783) |

Table 6 and 7 present the aggregated missing data imputation performance. The results in Table 6 and 7 demonstrate that the performance of MIM-RBFNN, utilizing the initial $\sigma$ assignment based on Equation 1 and employing common RBFs, is superior. Additionally, we observed that the initial $\sigma$ assignment using Equation 1 outperforms random $\sigma$ assignment in MIS-RBFNN using a single RBF for all datasets. This further accentuates the effectiveness of considering the time gap in RBF's receptive field for missing data imputation, as discussed in Appendix B.

