# OpenReview forum: "Time Series Missing Imputation with Multivariate Radial Based Function Neural Network"
_ICLR.cc/2024/Conference — Submitted to ICLR 2024_

### Official Review · Reviewer_Nj1D · 2023-10-27

**Soundness:** 2 fair
**Presentation:** 1 poor
**Contribution:** 2 fair
**Rating:** 1
**Confidence:** 4

**Summary:**

This paper offers a radial basis functions based imputation for time series data.

**Strengths:**

The paper tackles an issue that is relevant to multiple domains.

**Weaknesses:**

The current draft of the paper exhibits several weaknesses that require attention:

**Relevance:**
While the paper does tackle an important problem, it leaves ambiguity in how it addresses the weaknesses it points out in the existing literature. Most notably, the paper criticizes the assumptions made by existing strategies but doesn't explicitly state its own assumptions on the missingness process despite relying on either "missing at random" or "completely at random." Presently, the experiments only explore "missing completely at random," which raises questions about its practical applicability.

**Presentation:**
The description of the model is currently intricate and would benefit from a simplified presentation. It would be useful to clarify the problem as an optimization problem in which the role of the NN and RNN is clarified. Furthermore, the approach of modeling the residual of the previous model bears resemblance to a boosting approach but this connection is unclear.

**Significance:**
The paper mentions a few existing works but overlooks contributions related to Gaussian processes and RBF interpolation [1]. It is essential for the paper to describe this existing literature and clarifies how it distinguishes itself and builds upon it.

**Technical Soundness:**
As previously mentioned, the methodology needs clarification to assess its technical soundness. Moreover, the experimental setup should incorporate hyperparameter tuning to ensure that the observed performance differences are not solely due to inadequate parameter choices. Additionally, including confidence intervals in the results would help in understanding the significancne of the proposed strategy's improvement.

[1] Wright, G. B. (2003). Radial basis function interpolation: numerical and analytical developments. University of Colorado at Boulder.

**Questions:**

Several technical claims in the paper appear unjustified, the following needs clarification or validation:

- The use of 'Instability of deep learning' in the abstract is unclear.
- The statement "In healthcare, time series data are used for classifying patient outcomes" seems to oversimplify the role of time series data in healthcare.
- In the paragraph discussing complete case analysis, it should be mentioned that this technique not only introduces "analytical complexities" but is also biased under non-MCAR patterns. Furthermore, it affects the uncertainty in the analysis as fewer data points are available for model training.
- The claim that "However, the practical usability of GRU-D may be limited when there is no clear inherent correlation between the missing patterns in the dataset and the prediction task" requires justification. The paper should elaborate on why this might be the case and under what conditions GRU-D may not provide informative results.
- The paper should clarify what "GRUI" stands for.
- When stating that RBF networks are often referred to as universal function approximators, the paper should substantiate this claim with relevant references, such as "Park, J., & Sandberg, I. W. (1991). Universal approximation using radial-basis-function networks. Neural computation, 3(2), 246-257."

---

### Official Review · Reviewer_zQKi · 2023-10-29

**Soundness:** 3 good
**Presentation:** 1 poor
**Contribution:** 2 fair
**Rating:** 3
**Confidence:** 3

**Summary:**

The authors propose a deterministic time-series imputation method built upon RBFs, which provides a continuous approximation of the given time-series data. These continuous approximations are then incorporated into a bi-directional RNN architecture to enhance imputation performance. The method is evaluated on two datasets (with an additional dataset for ablation study only in the Appendix), demonstrating improved imputation performance over conventional deterministic imputation methods.

Overall, the paper is not ready for publication at ICLR in its current form. Firstly, the exposition of the paper is poor, with notations that are not well-defined and consistent throughout the document. Secondly, the paper's position is not clearly stated, and the technical contribution over previous work is vague. Thirdly, the learning of RBFs is not clearly described. Lastly, the experiments are limited to time-series data favorable for their design choices, and the technical claims are not well supported by the evaluation.

**Strengths:**

1. Continuous function approximation is relatively simple but can be powerful for certain types of time-series data.

**Weaknesses:**

1. The positioning of this work is weak; the motivation and contribution over many previous time-series imputation works are not clearly stated in the Introduction (paragraph 2) and in the Related Work.
2. Many related works based on generative models—such as VAE-based methods like GP-VAE [A] and diffusion model-based methods like CSDI [B]—that do not fall within the limitations stated in the Related Work section are missing. Moreover, the reasons for the confounding effects of the autoregressive imputation methods and why the proposed methods deviate from such limitations are not well clarified.
3. The procedure of learning multiple RBFs based on the residuals (in Section 3.1) needs improvement; currently, it is difficult to follow. Some notations are used without clear definitions.
4. The initialization of RBF components and how those components are trained based on the given time-series dataset are not clearly described.
5. Equation (11) should be clarified.
6. The experiments should provide a more in-depth analysis of the impact of different missingness assumptions (e.g., MAR, MANR) and a wider range of time-series datasets. Some important time-series imputation methods (e.g., GP-VAE [A], CSDI [B]) are missing in the comparison.
7. Evaluations on the ETT dataset are only given in the Appendix. It is not clear why the authors mainly focused their ablation study on a dataset not introduced in the manuscript.


[A] Fortuin et al., “GP-VAE: Deep Probabilistic Time Series Imputation,” AISTATS 2020.

[B] Tashiro et al., “CSDI: Conditional Score-based Diffusion Models for Probabilistic Time Series Imputation,” NeurIPS 2021.

**Questions:**

1. Regarding Weakness 2: The imputation of the proposed method is carried out in two phases: i) constructing RBFs and ii) using autoregressive RNN predictions using the continuous functions learned in phase i). Considering this, how can the authors claim that the proposed method deviates from the confounding effects of autoregressive models mentioned in Related Work?
2. The authors demonstrated in Appendix C that sharing the RBF functions across different features enhances imputation performance compared to using individual RBF functions. However, the reviewer believes that this approach could pose challenges when two time-varying features exhibit distinct temporal dynamics, such as different periodicities, a common occurrence in healthcare datasets (i.e., the domain mentioned in one of the applications). One can expect similar behaviors across different features for the evaluated datasets. What happens if such property is violated?
3. Some other time-varying features may be informative about a given time-varying feature, which, in turn, will help in imputing missing values in that feature. However, the proposed RBF construction lacks such dependencies. Please describe what happens if there exist strong dependencies across time-varying features.

---

### Official Review · Reviewer_vWr9 · 2023-10-31

**Soundness:** 3 good
**Presentation:** 2 fair
**Contribution:** 2 fair
**Rating:** 5
**Confidence:** 4

**Summary:**

The paper proposes an RNN based architecture for imputation of missing data in multivariate time series. The architecture relies on the Radial Basis Function Neural Networks to express the continuous representation of a multivariate timeseries with missing data points. Such imputed data-matrix is then fed to the RNN architecture, along with mask, incomplete time-series and time gap matrix. And trained by optimizing a sum of multiple losses, including the feature-based estimation loss, continuous function loss, history-based estimation loss, consistency loss. Incialization procedure for model parameters is also detailed, and training hyperparameters are provided. The proposed architecture was applied on few real world datasets and imputation performance (as measure by MEA and MRE) suggest improvement over several competitors.

**Strengths:**

Detailed description of this rather complex architecture makes the paper easy to follow.

The empirical results suggest improvement over the competitor approaches, and in the case of Human Activity dataset both MIM-RBFNN component, as well as the complete MIRNN-CF model beat the competition in the imputation task across the range of missing ratio (up to 80%).

It is also appreciated that ablation study is provided - in a sense that from Table 1 it is visible how much performance is attributed to MIM-RBFNN, and what is the improvement further achieved by MIRNN-CF.

**Weaknesses:**

Although paper is generally well written, some claims I would disagree with. For example, as a part of motivation it was stated that some competitor approaches "they often rely on strong assumptions, such as presuming that the data adheres to a particular model distribution, or they incorporate information beyond what is available in multivariate time series data, such as specific location data. Consequently, these approaches may struggle to capture the data distribution adequately and may not guarantee good performance on data that deviates from the model assumptions or lacks location information". It is fair statement though, but it would be fair to also recognize that the Radial Basis Function also assume 'strong assumption' that the value of a function depends only on the distance from a certain fixed point. Moreover Gaussian RBF is using additional strong assumption that value of function decays exponentially as the distance increase.


Also, empirical evaluation, while impressive, it appears to be conflicted with some previously published results. For example, as reported in SAITS paper its performance should be comparable to BRITS on AirQuality dataset - but in Table 1 it is significantly worse than BRITS.
Also self-reported results on AirQuality application (in the BRITS paper) appear better than reported in Table 1.

Its not clear why MIM-RBFNN trains a sequence of RBFNNs in a boosting-like fashion. It would be good to make comparison against using just a single RBFNN but with more parameters (more expressive). Also, comparison against Gaussian Processes would be interesting.

**Questions:**

In Baseline section, MICE is mentioned as one of 8 baseline models. However it is not in the Table 1. Are the results in table from MF or from MICE?

---

### Meta-Review · Area_Chair_kSU5 · 2023-12-03

**Metareview:**

The paper proposes a Missing Imputation Multivariate RBFNN (MIM-RBFNN), for addressing missing data imputation in time-series data. The model is based on RBF for learning local information to learn continuous information. This is also integrated in a recurrent framework for capturing temporal information.

While empirical evaluations show that the proposed method outperforms SOTA, there are many weaknesses in the paper.

The manuscript makes some strong unsubstantiated claims about existing literature. Although strong assumptions in the existing literature are identified, it is unclear how the method proposed in this paper addresses these assumptions (Reviewer Nj1D, Reviewer vWr9). There are notational discrepancies (Reviewer Nj1D, Reviewer zQKi) and the related works are not complete (Reviewer Nj1D, Reviewer zQKi). In summary, the manuscript is not complete, as noted by the reviewers.

**Justification For Why Not Higher Score:**

The manuscript makes some strong unsubstantiated claims about existing literature. Although strong assumptions in the existing literature are identified, it is unclear how the method proposed in this paper addresses these assumptions (Reviewer Nj1D, Reviewer vWr9). There are notational discrepancies (Reviewer Nj1D, Reviewer zQKi) and the related works are not complete (Reviewer Nj1D, Reviewer zQKi). In summary, the manuscript is not complete, as noted by the reviewers.

**Justification For Why Not Lower Score:**

N/A

---

### Decision · Program_Chairs · 2024-01-16

Reject